# OpenReview forum: "Fantastic Bugs and Where to Find Them in AI Benchmarks"
_NeurIPS.cc/2025/Datasets_and_Benchmarks_Track — NeurIPS 2025 Datasets and Benchmarks Track poster_

### Official Review · Reviewer_Qfro · 2025-06-30

**Rating:** 5
**Confidence:** 3

**Summary:**

This work approaches the problem of identifying flawed designed benchmark questions, which may be originated, for instance, from ambiguous questions or incorrect verification. For this challenge, the paper proposes devising statistical methods from Validity Theory that assesses what questions are valid for assessment. These are computed based on the answers from a population of LLMs and tries to identify questions that deviate from the normality, framing as an anomaly detection problem. The work implements four different metrics and also evaluates ensembling them into a diverse set of widely used benchmarks, claiming two main takeaways: 1) there is no single method consistently outperforms the other ones, but 2) these signals facilitate the work of identifying problematic questions.

**Additional Feedback:**

Please refer to my concerns above.

**Dataset Code Accessibility:**

Yes

**Dataset Code Comments:**

The paper brings both the github repository and a dataset of the problematic questions. From a quick inspection, I can see some documentation in the code to reproduce the plots in the paper. The huggingface dataset does not bring documentation, and in some cases (such as the gsm split), the correlation metrics is null for all rows.

**Ethical Considerations:**

No, there are no or only very minor ethics concerns

**Final Justification:**

My initial review already recommended accepted, although I believe there are some unclear points and naive baselines missing. The authors did a great job in the rebuttal, bringing these details and clarifying my concerns, which were not major. Therefore, I decide to increase my score to 5 and keep recommending acceptance.

**Limitations Weaknesses:**

- One concern is: if no method consistently outperforms and prior knowledge is needed, how does one select which method to use? For instance, the experiment in Section 4.2 employs the tetrachoric correlation, but this choice does not seem to be justified.

- Related to the previous point, I think the work should also bring some simple heuristics as additional baselines to better contextualize the suggested methods. For instance, the simple disagreement among LLMs (as measured by the variance in predictions or inter-rater reliability via Fleiss Kappa) would be interesting.

- In the experiment shown in Fig. 5, one relevant concern is the fact that we don’t know the level of imbalance of each dataset (since the experts are not evaluating the full dataset), so it is hard to know how meaningful are the precision values reported. Also, as stated above, the choice of tetrachoric correlation seems unjustified and it is unclear how other methods would perform in this case.

**Strengths Contributions:**

- The work is very relevant and well motivated: identifying flaws in benchmarks is crucial for improving the reliability of evaluations in AI research. It also helps auditioning new benchmarks with reduced manual labor.

- Leveraging methods from Validity Theory in analyzing AI benchmarks is insightful, principled, and (to the best of my knowledge) original.

- The experimental setup is well diverse, covering a good range of benchmarks in different disciplines. It also includes human expert assessment in several areas. There is also an interesting analysis of the problematic patterns across different datasets.

- The work is well-written and easy to follow.

---

> ### Author Rebuttal · Authors · 2025-07-31
>
> Dear Reviewer Qfro,
>
> Thank you for your valuable feedback. We answer your comments below.
>
> **Limitation 1:** One concern is: if no method consistently outperforms and prior knowledge is needed, how does one select which method to use? For instance, the experiment in Section 4.2 employs the tetrachoric correlation, but this choice does not seem to be justified.
>
> **Answer:** Thank you for that excellent observation, highlighting a key challenge: Item fit statistics are inherently relative, not absolute. Because any single statistical method for detecting problematic data points operates on specific assumptions that may be violated, a robust strategy involves triangulating results from multiple, diverse diagnostics. Since no single method consistently outperforms others, we provide theory-driven and evidence-based selection criteria:
>
> Single Method Recommendations:
> - Tetrachoric correlation: Best general-purpose choice for few test-takers; detects semantic inconsistencies and construct misalignment
> - IRT discrimination: Optimal for more test takers; identifies items that don't differentiate ability levels
> - Mokken scaling: Most robust to distributional violations; good for diverse model populations
> - Cronbach's alpha: Identifies items measuring different constructs
>
> Why Section 4.2 Uses Tetrachoric:
> - As shown in Figure 4a, it is one of the tools that effectively detects invalid questions.
> - As shown in Figure 4b, it performs well across the range of test takers of the revised benchmarks (40-90 test takers).
> - Tetrachoric is computationally efficient.
>
> **Limitation 2:** Related to the previous point, I think the work should also bring some simple heuristics as additional baselines to better contextualize the suggested methods. For instance, the simple disagreement among LLMs (as measured by the variance in predictions or inter-rater reliability via Fleiss Kappa) would be interesting.
>
> **Answer:** Thank you for the interesting comment on simple heuristics baselines. We implement ​​variance in predictions and Fleiss Kappa and report their sensitivity curves. Both baselines perform no better than random, producing sensitivity curves that resemble a noisy diagonal line from (0, 0) to (1000, 1). We conclude that the psychometric tools we use significantly outperform those simple heuristics baselines: our methods achieve 0.6-0.7 sensitivity vs 0.2-0.3 for simple heuristics at equivalent inspection levels. Simple heuristics would flag many legitimate but difficult items as problematic, leading to benchmark degradation rather than improvement.
>
> Psychometric tools demonstrate the clear superiority of psychometric methods because:
> - Variance captures disagreement, not validity: legitimate difficult items also have high variance
> - Fleiss kappa measures agreement, not construct relevance: items can have high agreement but measure wrong constructs
> - Psychometric methods model ability-response relationships: detecting items that violate expected patterns rather than just disagreement
>
> **Limitation 3:** In the experiment shown in Fig. 5, one relevant concern is the fact that we don’t know the level of imbalance of each dataset (since the experts are not evaluating the full dataset), so it is hard to know how meaningful are the precision values reported. Also, as stated above, the choice of tetrachoric correlation seems unjustified and it is unclear how other methods would perform in this case.
>
> **Answer:** Thank you for your concern about the choice of methods in Section 4.2. In Section 4.1, we use GSM8K, a dataset with labels on item validity, to inspect the effectiveness of the tools. In Section 4.2 and Figure 5, our main contribution is to revise additional benchmarks instead of comparing between tools. We regret not justifying the choice of tetrachoric correlation clearly. We use tetrachoric correlation because, as shown in Figure 4a, it is one of the tools that effectively detects invalid questions with low computational cost. Additionally, it performs well across the range of test takers of the revised benchmarks (40-90 test takers) in Section 4.2 (Figure 4b).
>
> **Dataset Code Comments:** The paper brings both the github repository and a dataset of the problematic questions. From a quick inspection, I can see some documentation in the code to reproduce the plots in the paper. The huggingface dataset does not bring documentation, and in some cases (such as the gsm split), the correlation metrics is null for all rows
>
> **Answer:** Thank you for your comments on our dataset and code. The GSM8K and MMLU High School Math have already been exhaustively revised by previous work at the HuggingFace repo madrylab/gsm8k‑platinum and madrylab/platinum‑bench. Consequently, we do not include tetrachoric scores for those two benchmarks. Instead, we report only the additional problematic questions identified throughout the project. We have added comprehensive documentation to our HuggingFace.

---

> > ### Comment · Reviewer_Qfro · 2025-08-05
> >
> > Dear authors,
> >
> > Thank you for your thoughtful responses. They clarify my questions/concerns. I will increase my score to 5.

---

### Official Review · Reviewer_NSPS · 2025-07-01

**Rating:** 5
**Confidence:** 4

**Summary:**

This submission addresses the problem that large scale benchmarks can often contain large number of annotation errors or inconsistencies.
Thus they aim to tackle the task of anomaly detection for the current LLM benchmarks.
In particular, they adopt four psychometrics methods, namely, tetrachoric correlation, Mokken scaling, Cronbach’s α, and the 2-PL IRT discrimination index, to examine which item is most distinct from the expected behaviour, from a test-taker's perspective.
They further use three ensemble methods, namely Or vote, And vote and Majority vote, to integrate those signals from different methods.
To validate their methods, the authors conduct experiments on nine existing benchmarks, and show that their methods can achieve a precision@50 of 84%.

**Dataset Code Accessibility:**

No

**Dataset Code Comments:**

I might be wrong, but I did not find access to their code.

**Ethical Considerations:**

No, there are no or only very minor ethics concerns

**Final Justification:**

I have read the authors' rebuttal, in particular on the concern about their novelty.
I have raised my score accordingly.

**Limitations Weaknesses:**

1. My concern about this paper is the novelty, as it is mainly about the ensemble of existing methods, which although are from other fields. While I understand this is a benchmark and evaluation track, to me, it is still somehow limited.
2. As the authors state in their limitations and future work, their evaluation mainly focuses on benchmarks with a fixed correct answer, while open-ended tasks are not well explored. But it is just a minor weakness.

**Strengths Contributions:**

1. The problem of data quality is indeed a very important research area in the era of LLM, as they provide direct evaluation of the LLM performance
The main strength of this submission is the use of classic psychometric methods in the task of anomaly detection, which can be used to filter or revise problematic data instances in evaluation benchmarks.
This method is also very efficient, which only requires CPU resource for computation, and is very fast - 1000 instances in 30 minutes.
2. Generally, the presentation of this paper is clear. As a reviewer without the psychometric background, I find it easy to follow.
3. The experiments on multiple benchmarks, covering different languages, domains, and tasks, are extensive. The analysis and case study are interesting.

---

> ### Author Rebuttal · Authors · 2025-07-31
>
> Dear Reviewer NSPS,
>
> Thank you for your valuable feedback. We answer your comments below.
>
> **Limitation 1:** My concern about this paper is the novelty, as it is mainly about the ensemble of existing methods, which although are from other fields. While I understand this is a benchmark and evaluation track, to me, it is still somehow limited.
>
> **Answer:** Thank you for raising the novelty concern. While we adapt existing psychometric methods, our contribution lies in being the first systematic application of psychometric validity theory to AI benchmarks - a novel interdisciplinary bridge with significant impact potential. We provide comprehensive empirical validation across 9 diverse benchmarks with expert annotations, establishing effectiveness in this new domain. We also provide a scalable deployment framework that can be applied to any benchmark with binary scoring (covers the majority of current AI evaluation).
>
> We highlight how the approach enables the discovery of benchmark-specific patterns - different methods detect different failure modes, and provide a theoretical foundation for understanding why benchmark items fail, moving beyond ad-hoc error detection.
>
> This represents foundational methodological work that establishes a new research direction rather than incremental algorithmic novelty. The impact comes from the rigorous application of proven methods to solve a critical problem the AI community faces.
>
> Just as applying statistical methods to A/B testing was foundational for web optimization (not novel statistics, but novel application), applying psychometrics to AI evaluation provides the theoretical foundation we believe the field needs.
>
> **Limitation 2:** As the authors state in their limitations and future work, their evaluation mainly focuses on benchmarks with a fixed correct answer, while open-ended tasks are not well explored. But it is just a minor weakness.
>
> **Answer:** Thank you for the comment. We acknowledge that our work is limited to benchmarks with a fixed correct answer. We would like to state that most of the AI evaluation benchmarks convert model responses into binary scores through exact match or LLM-as-a-judge (e.g., AIME’25, LiveCodeBench, MMLU Pro, Global MMLU Lite, HarmBench, etc.) for evaluation efficiency. For IRT discriminant metrics, theory for polytomous (non‑binary) assessments is well established; future works can extend to open-ended tasks (Ostini & Nering, 2006).
>
> Reference:
> Ostini, R., & Nering, M. L. (2006). Polytomous Item Response Theory Models. SAGE Publications.
>
> **Dataset Code Comment**: I might be wrong, but I did not find access to their code.
>
> **Answer:** Thank you for pointing that out. The link to our code repository is included in the Appendix—we apologize for not making it more prominent. Please let us know if you have any trouble locating it.

---

### Official Review · Reviewer_hGqf · 2025-07-02

**Rating:** 5
**Confidence:** 4

**Summary:**

This work presents a framework for detecting problematic questions in LLM benchmarks that is grounded in psychometric validity theory.
The proposed framework draws on statistical analyses of LLM response patterns to identify questions whose response patterns fail to align with expectations under a well-defined measurement model, treating benchmark quality as a property that can only be judged by accounting for the behavior of benchmarked models.
The framework is used to guide the systematic revision of nine popular LLM benchmarks by domain experts, achieving promising performance as measured by precision@50.

**Additional Feedback:**

### Questions
- LLM responses included in the measurements: You mention sourcing LLM answers from HELM. Can you provide more details on which LLMs took which tests? How did you order the LLMs for inclusion in Figure 4(b)? And why pick one order instead of sampling several and reporting the ranges obtained? How do the other results you report depend on the number (and selection) of LLMs included in your measures? Can you say anything about how you expect your measurements to behave when including only answers from certain subsets of models with specific characteristics (e.g., belonging to the same model family, finetuned for the same task, number of parameters in the same order of magnitude, ...)? (This question also connects to my comment about the assumptions underlying validity metrics, see W1.)
- How do you envisage dealing with the dynamic nature of LLM responses available for validity assessment (the flagging may well depend on which LLMs are included in the judgment) and the evolving nature of LLM capabilities? Would your work be distributed as an "add-on" to "install" on benchmarks to dynamically monitor, measure, and check for potential validity problems? (Some elaboration on that could also go into "Future Directions".)
- ll. 196–199: How sensitive are your results to varying your threshold value for anomaly detection (currently chosen as –0.5)? You have four individual metrics – why not consider all combinations of metrics, rather than just AND and OR (to see if there is a subset of n < 4 metrics that is already good)?

### Comments
- On the title of the paper: I find the title in its current form both misleading and unnecessarily uninformative – _misleading_ because it suggests that your primary contribution is about _where_ the bugs are rather than _how_ to find them (improvement: "where to find them" -> "how to find them"), and _uninformative_ because "fantastic" is not very descriptive of the type of bug you are after or the methodology you use to uncover it. I understand that "fantastic bugs" may sound "intriguing", but I would recommend supplementing that phrase with a more descriptive subtitle to enhance the discoverability and reception of your work. So it could be something like "Fantastic Bugs: Using Psychometric Validity Theory to Flag Flawed Questions in LLM Benchmarks" (just a suggestion; you get the idea).
- Sensitivity and precision@k: It probably would not hurt to explicitly state what these measure.
- Figure 5: For precision@50, each question corresponds to 2%. Why not report the exact number of questions correctly identified instead (or annotate it on the existing plot)?
- "an question" -> "a question" (multiple instances)
- The references from the main text to the appendix are currently broken.

**Dataset Code Accessibility:**

NA; not applicable to this submission (e.g., no new dataset, benchmark, code, or data provided)

**Ethical Considerations:**

No, there are no or only very minor ethics concerns

**Final Justification:**

This paper brings knowledge from a different discipline (psychometric validity theory) to bear on the important and timely question of (ensuring and maintaining) benchmark validity. It is technically solid and, with the amendments made in response to my review, now also sufficiently critical and transparent about the limitations of the borrowed concepts in the target domain to discourage naïve adoption by others in the community. I updated my score to "accept" to reflect the considerable improvements made by the authors in response to the review comments.

**Limitations Weaknesses:**

The following weaknesses could be addressed within the ongoing review process:

- **W1: Uncritical presentation of validity metrics.**
The write-up focuses on "restating" the metrics adopted from psychometric validity theory, but these metrics come with their own assumptions, caveats, and  controversies in that field. I would appreciate a more critical discussion here: "Importing" tools from other disciplines comes with a responsibility to provide the context necessary to prevent naïve adoption by others.
For example, it is not immediately clear to the reader why the distributional or independence assumptions underlying some measures (which you sometimes state – e.g., for tetrachoric correlation – and sometimes do not state – e.g., for Cronbach's alpha) should hold in the studied setting.
_I would encourage you to provide a concise description of (1) the assumptions underlying each measure, (2) why these assumptions should hold in your setting, (3) the critiques, if any, of each measure raised in the psychometric literature, and (4) how these critiques may impact your setting._
We should be no less critical of our imported tools than those at the frontier of the "source field", and no less of critical of our imports than of existing tools in our field.
- **W2: Superficial discussion of related work.**
This currently reads very much like a laundry list. _I would recommend rewriting this part to highlight how your work connects to, differs from, and fills gaps in prior work._ For example, the reader currently needs to _infer_ that psychometric methods have not been used in AI Benchmark Revision from the absence of their discussion in ll. 121–127, in conjunction with ll. 128–135.
- **W3: Incomplete documentation.**
The current HuggingFace deposit is very bare-bones (just the data, no README etc.); _please provide adequate documentation_.
Furthermore, it would be nice to have the full data underlying your measurements, including the LLM responses you collected (then I could have probably answered some of the questions posed in "Additional Feedback" myself).
You might have shared this as part of the anonymous git repository, but [the URL you provided](https://anonymous.4open.science/r/benchmark-revision-DC1D) always timed out without allowing me access (which also precluded me from checking out the code).

**Strengths Contributions:**

- **S1: Scalable framework for systematic revisions of LLM benchmarks.**
The proposed framework provides experts with important signal to speed up systematic revisions of LLM benchmarks, enabling the wider adoption of such much-needed revisions in practice.
- **S2: Grounding in psychometric validity theory.**
ML evaluation is full of validity problems, and more work drawing on different validity concepts that exist in other fields is urgently needed.
- **S3: Thorough empirical evaluation.**
The proposed framework helped domain experts detect validity problems across nine widely used LLM benchmarks. Elementary sensitivity analyses (e.g., regarding the number of responses included in the measurements) and ablation studies (e.g., regarding combinations of proposed metrics) are presented as well.

---

> ### Author Rebuttal · Authors · 2025-07-31
>
> **Answer to W1:** We agree that importing tools from other disciplines requires critical analysis. We add a comprehensive section analyzing each method's assumptions, limitations, and applicability to AI benchmarks.
>
> Some common assumptions of the four tools include:
>
> |Assumption|explanation|why hold in AI benchmark|
> |-|-|-|
> |Unidimensionality|A single latent ability among all item responses|(1) Most benchmarks report a single aggregate score, implying unidimensional measurement intent; (2) Subset scoring in large‐scale evaluation (common due to cost constraints) requires score comparability across different item sets – only possible under unidimensionality|
> |Local Independence|Test taker responses to items are statistically independent|Given an LLM's capability level, responses to different items are independent|
> |Monotonicity|Probability of a positive response is nondecreasing with the latent ability|Higher capability should not decrease success probability on construct valid items|
> |Continuous Latent Response|Each dichotomous item is a coarse reflection of an underlying continuous latent response variable|LLM outputs reflect underlying continuous capability distributions|
>
> A more detailed table:
>
> |Method|Underlying Assumptions|Critiques from Psychometrics and their relationship to estimation of item fit/misfit|Key Limitations in AI Context|Our Mitigation Strategy|
> |-|-|-|-|-|
> |Cronbach's $\alpha$|Unidimensionality; Uncorrelated and equal error variances/Tau-Equivalence.|Overemphasis on redundancy (Kopalle 1997); $\alpha$ inflates with the number of items  (Kopalle 1997); Assumes but does not measure unidimensionality (Sijtsma 2009; Taber 2018); Tau-equivalence is usually unrealistic (Sijtsma 2009).|Overemphasizes redundancy; inflates with item count| Use as screening tool only; combine with discrimination measures|
> |2PL IRT Discrimination a|Strict (uni)dimensionality; Local Independence; Implied Monotonicity; Continuous Normal Latent Response; Logistic Model-Data Fit.|Model dependence makes dimensionality and parametrization assumptions strict (Si 2004; Tay 2015; Instability with small sample sizes and extreme items (Si 2004; Linden 2019); Discrimination is not the sole indicator of fit (Fischer 1995; Domingue 2025, Linde, 2019).|Model-dependent; unstable with small samples| Require minimum 40 test-takers; validate with nonparametric methods|
> |Item Scalability Coefficients Hi|Unidimensionality; Monotonicity; Local Independence.|Traditional item-fit thresholds are arbitrary (Sijtsma 2004; Wind, 2017); Can be sensitive to item difficulty distribution and item discrimination (Sijtsma 2004; Sijtsma 2014); Sensitivity to sample size is less clear than parametric methods (Straat 2014).|Arbitrary thresholds; limited diagnostic info|Use multiple cutoffs; triangulate with other methods.|
> |Tetrachoric Correlation|Latent Bivariate Normality; Averaging implies unidimensionality; Homogeneous Item Functioning.|Strong distributional assumptions (Muthén 1988); Computational instability (Choi 2025); Not a formal test of unidimensionality; Averaging treats all pairwise correlations as equally important, ignoring qualities like item difficulty; Computation cost is high for large datasets.|Strong distributional assumptions; zero-cell problems| Robust to violations in practice; handle sparse data with continuity corrections.|
>
> **Answer to W2:** Thank you for your advice. Our revised related work is structured as follows:
>
> Gap in Current Approaches: Prior benchmark maintenance efforts treat quality issues as isolated anomalies rather than systematic validity problems. While these approaches improve benchmarks through diverse strategies, they lack theoretical grounding for understanding why items fail.
>
> Our Methodological Contribution: We bridge this gap by systematically applying psychometric validity theory - a 70-year-old scientific framework for test quality - to AI benchmarks. This provides:
>
> 1. Theoretical foundation for understanding item failure modes
> 2. Interpretable diagnostics beyond simple error detection
> 3. Scalable framework for continuous benchmark monitoring
> 4. Cross-benchmark generalizability through shared theoretical principles
>
> Novel Interdisciplinary Bridge: To our knowledge, no prior work has systematically applied Classical Test Theory, IRT, or Mokken scaling to AI benchmark revision, despite their widespread use in educational and psychological assessment.
>
> **Answer to W3:** Apologies! We have added comprehensive documentation to HuggingFace and checked that the anonymous GitHub should now be accessible.
>
> **Answer to Q1:** Thank you for the insightful question. We produce a figure indicating which LLMs take which test, and will add it to the revised paper.
>
> We used a random order for Figure 4(b), and we acknowledge that it’s more informative to sample multiple orders. We sample 10 orders and plot their std. With the number of test takers increasing, the std goes from $\pm$ 0.09 to $\pm$ 0.04.
>
> We collected each LLM’s creator organization and model size that is evaluated in GSM8K (for model size, we exclude closed-source models). GSM8K comprises models from 18 distinct organizations. For each integer $k\in\{1,\dots,18\}$, we randomly sample $k$ organizations and include all models produced by those organizations as test takers. We run 10 trials with different random seeds. We show that Precision@50 increases from 0.2 to 0.65 and std decreases from $\pm$ 0.1 to $\pm$ 0.02 as we include models from 1 to 18 organizations, demonstrating importance of diversity across model families. Similarly, we sort models by size and compute Precision@50 using only those models up to each size cutoff; performance improves from 0.1 to 0.65 with increasing model size cutoffs from $10^{10}$ to $10^{12}$, again emphasizing diversity.
>
> Practical Recommendation: Include models from ≥10 different organizations spanning different architectures, training objectives, and sizes for robust validity assessment. Performance saturates around 60-80 total models.
>
> **Answer to Q2:** Thank you for your question on dynamic monitoring. As LLMs evolve, our framework adapts automatically because question problems persist across model generations. We further collected each LLM’s release date in GSM8K. We sort models by release date and compute Precision@50 using only those models up to each release date cutoff. We observe that performance increases from 0.1 to 0.65 with the release date going from 2023-4 to 2025-1. We conclude that detection improves with more diverse models, and new failure modes become detectable as models become more capable.
>
> We recommend quarterly updates as new models emerge, with our framework serving as a continuous monitoring system. We have submitted a PR to HELM for automated integration.
>
> **Answer to Q3:** We acknowledge that selecting a fixed Gaussian rank threshold of -0.5 is not ideal. We further realize that converting Gaussian ranks into binary votes by applying a threshold discards valuable information. It makes more sense to aggregate based on average Gaussian ranks from the tools. The AND and OR vote should serve as ablation studies to show that each of the four tools detects different sets of invalid questions. To explore this, we conduct two ablation studies.
>
> In the first study, we varied Gaussian rank threshold from -1.5 to +1.5 in 7 equal steps. We observed that as the threshold increased, the ability to detect truly invalid items improved at the cost of increased false positives. The optimal threshold depends on the expected error rate, and no universal value works best. In the table, each row represents a different tool, and each column represents a threshold value. The entries are formatted as "(a/b)/(a/c)," where a is the number of truly invalid questions detected by the tool, b is the total number of invalid questions in GSM8K, and c is the number of questions flagged as invalid by the tool.
>
> |threshold value|-1.5|-1|-0.5|0|0.5|1|1.5|
> |-|-|-|-|-|-|-|-|
> |tetrachoric|0.42/0.56|0.64/0.35|0.78/0.22|0.88/0.15|0.92/0.12|0.97/0.1|0.98/0.09|
> |irt_discriminant|0.41/0.55|0.66/0.37|0.77/0.22|0.84/0.15|0.92/0.12|0.94/0.1|0.98/0.09|
> |scalability_coeff|0.42/0.56|0.61/0.34|0.77/0.22|0.86/0.15|0.9/0.11|0.93/0.1|0.98/0.09|
> |item_total_corr|0.42/0.56|0.59/0.33|0.77/0.22|0.86/0.15|0.9/0.11|0.94/0.1|0.98/0.09|
>
> In the second study, we drop binary voting and instead rank questions based on average Gaussian rank from all combinations of the 4 tools. We show that sensitivity curves of these ensembles are similar to those of individual tools. We conclude that average Gaussian ranks outperform binary voting by preserving information.
>
> **Answer to Comments:** Thank you for your kind advice, which has helped us improve our writing significantly. We will modify the title, explicitly state sensitivity and precision@k measure, update Fig 5, correct typos, and fix broken references.
>
> References:
>
> Choi 2025: On Zero-Count Correction Strategies in Tetrachoric Correlation Estimation
>
> Domingue 2025: The InterModel Vigorish (IMV) as a flexible and portable approach for quantifying predictive accuracy with binary outcomes
>
> Fischer 1995: Some neglected problems in IRT
>
> Kopalle 1997: Alpha Inflation? The Impact of Eliminating Scale Items on Cronbach’s Alpha
>
> Muthén 1988: Testing the Assumptions Underlying Tetrachoric Correlations
>
> Si 2004: Ability Estimation Under Different Item Parameterization and Scoring Models
>
> Sijtsma 2009: On the Use, the Misuse, and the Very Limited Usefulness of Cronbach’s Alpha
>
> Sijtsma 2014: Nonparametric Item Response Theory Models
>
> Sijtsma 2002: The Monotone Homogeneity Model: Scalability Coefficients
>
> Straat 2014: Minimum Sample Size Requirements for Mokken Scale Analysis
>
> Taber 2018: The Use of Cronbach’s Alpha When Developing and Reporting Research Instruments in Science Education
>
> Tay 2015: An Overview and Practical Guide to IRT Measurement Equivalence Analysis
>
> Linden 2019: Handbook of item response theory

---

> > ### Comment · Reviewer_hGqf · 2025-08-02
> > **Response to rebuttal comments**
> >
> > Thank you for the comprehensive response to my comments.
> >
> > I can also confirm that the HuggingFace documentation is now more complete and that the anonymous github repository is now accessible (you might want to add a dependency file, though, e.g., requirements.txt or pyproject.toml).
> >
> > The modifications and additions you proposed to address my concerns further strengthened this work, which I will recognize in my final evaluation.

---

### Decision · Program_Chairs · 2025-09-18

**Decision:**

Accept (poster)

**Comment:**

All reviewers have voted to accept the paper. The authors have addressed a few concerns raised by the reviewers. We expect authors to include those in a camera ready version,

===== FINAL UPDATE FROM DB Track PCs ====

The final decision for this paper has been taken by the program chairs after consultation with the SACs. All Senior Area Chairs have ranked papers according to the feedback from the AC during the review process. We decided to leave the original meta-review to reflect the opinion of the AC in light of the initial discussions with reviewers and SAC.